

# Fast and automatic estimation of transition state structures using tight binding quantum chemical calculations

Maria H. Rasmussen and Jan H. Jensen

Department of Chemistry, University of Copenhagen, Copenhagen, Denmark

## ABSTRACT

We present a method for the automatic determination of transition states (TSs) that is based on Grimme's RMSD-PP semiempirical tight binding reaction path method (J. Chem. Theory Comput. 2019, 15, 2847–2862), where the maximum energy structure along the path serves as an initial guess for DFT TS searches. The method is tested on 100 elementary reactions and located a total of 89 TSs correctly. Of the 11 remaining reactions, nine are shown not to be elementary reactions after all and for one of the two true failures the problem is shown to be the semiempirical tight binding model itself. Furthermore, we show that the GFN2-xTB RMSD-PP barrier is a good approximation for the corresponding DFT barrier for reactions with DFT barrier heights up to about 30 kcal/mol. Thus, GFN2-xTB RMSD-PP barrier heights, which can be estimated at the cost of a single energy minimisation, can be used to quickly identify reactions with low barriers, although it will also produce some false positives.

## INTRODUCTION

The computational determination of chemical reaction networks (*Maeda, Ohno & Morokuma, 2013*; *Bhoorasingh et al., 2017*; *Kim et al., 2018*; *Unsleber & Reiher, 2020*; *Robertson, Ismail & Habershon, 2019*; *Suleimanov & Green, 2015*) requires that the estimation of barrier heights and/or location of transition states (TSs) be automated. Many methods for automated barrier height estimation and TS location have been proposed (*Mills & Jónsson, 1994*; *Jónsson, Mills & Schenter, 1995*; *Henkelman, Uberuaga & Jónsson, 2000*; *Weinan, Ren & Vanden-Eijnden, 2002*; *Weinan, Ren & Vanden-Eijnden, 2005*; *Peters et al., 2004*; *Zimmerman, 2013a*; *Bhoorasingh et al., 2017*; *Schlegel, 2011*). However, the computational demand of these methods are significantly higher than for locating minima.

Recently, *Grimme (2019)* presented a method (root mean square deviation-push-pull or RMSD-PP) for the rapid estimation of reaction paths based on a semiempirical tight-binding model (GFN2-xTB (*Bannwarth, Ehlert & Grimme, 2019*; *Grimme, Bannwarth & Shushkov, 2017*)). The predicted path can be used in a barrier estimate and the maximum energy structure as a TS guess in more expensive methods. Here, the performance of both are tested. This method is attractive to use when screening large amounts of reactions, as

Corresponding author
Jan H. Jensen, jhjensen@chem.ku.dk

it is not much more expensive than a geometry optimization and the GFN2-xTB method has been parameterised for the entire periodic table up to $Z = 86$. However, for it to be practically useful it needs to work in an automated framework.

Furthermore, we investigate whether the RMSD-PP reaction path can be used to distinguish reactions that have high and low barriers at the DFT level. If so, the RMSD-PP method could be used to increase the efficiency of the high throughput determination of reaction networks, where one is usually interested in relatively low-energy barriers.

The paper is organized as follows. First, the automated procedure for locating transition states is presented. Then, the method is tested on 100 elementary reactions suggested by *Zimmerman (2013a)* and *Zimmerman (2013b)*. Next we test whether the DFT barrier heights can be estimated using the GFN2-xTB RMSD-PP reaction path and test some commonly used methods for validating transitions states. Finally, we summarize our conclusions.

## METHOD

The idea behind the RMSD-PP method is to add a Gaussian biasing potential to the electronic energy ($E_{tot}^{el}$) "pushing" the molecule away from the reactant structure and a Gaussian biasing potential "pulling" the molecule towards the product structure.

$$E_{\text{tot}} = E_{\text{tot}}^{\text{el}} + k_{push}e^{-\alpha\Delta r^2} + k_{pull}e^{-\alpha\Delta p^2} \tag{1}$$

Here $k_{push} > 0$ and $k_{pull} < 0$, and $\Delta_r$ and $\Delta_p$ are the RMSDs between the current structure and the reactant and product, respectively. A geometry optimisation with this energy function is performed starting with the reactant structure and the geometry of each optimisation step is saved, re-optimised with three steps without the biasing potentials, and $E_{\text{tot}}^{\text{el}}$ recorded. All these structures (typically 30–200) and the corresponding energies represent the reaction path and the associated computational cost thus corresponds to that of a geometry optimisation. Representative timings are shown in Table S1.

Figure 1 shows a flowchart of the automated procedure for locating transition states (TSs). The reactant and product structures with same atomic ordering are required as input. The procedure starts with an RMSD-PP path search run with respective $k_{pull}$ and $k_{push}$ values of −0.02 and 0.01 Hartrees ($E_h$) and an $\alpha$ of 0.6 Bohr $^{-1}$ ($1/a_0$) (parameter set 1, Table S2). In addition to this run, two additional runs are performed where the $k_{pull}$ and $k_{push}$ values are multiplied by 1.5 and 2.25. A run is deemed successful if the root mean square deviation (RMSD) of the end structure compared to the product structure is less than 0.3 Bohr and the reaction path with the smallest absolute values of $k_{pull}$ and $k_{push}$ is selected. If the reaction does not complete, the setup for the path search is changed: the last structure of the run is saved and used as product structure in the next run while the product structure is used as reactant structure (trial 2, parameter set 1, Table S2). The same procedure is then repeated for trials 3–5 (Table S2) until completion is achieved. If all five attempts fail, then the entire procedure is repeated with an electronic temperature of 6,000 K (increased from 300 K). If the reaction again fails to complete then the method is deemed to fail for the reaction, although we did not observe this for the reactions considered in

this paper. We also test a slightly different parameter set (parameter set 2, Table S2), where $k_{push}$ is lowered to 0.008 $E_h$ for the first try.

Once the reaction has completed and the path found, the maximum energy structure along the path is extracted along with the two neighbouring structures. A linear interpolation (10 points from maximum energy structure to both neighbours) is performed and the interpolated structures are subjected to single point energy calculations using both Density Functional Theory (DFT) and GFN2-xTB. All DFT calculations are performed with the Gaussian 16 program (*Frisch et al., 2016*). The maximum GFN2-xTB energy along the interpolated path is used to estimate the GFN2-xTB barrier (orange part of the flow chart, Fig. 1). The maximum energy structure based on DFT calculations is used as initial guess for the TS structure in a DFT TS search using the Berny optimization algorithm (*Schlegel, 1982*) [*opt=(calcall, ts, noeigen)*]. Whether the correct TS is found is evaluated based on an intrinsic reaction coordinate (IRC *Fukui, 1970*) path search in both forward and reverse direction from the found TS. From the endpoint structures of the IRC, the adjacency matrices are extracted. The adjacency matrix for an $N$ atom system is an $N \times N$ matrix with 1 on the off-diagonal elements linking atoms that are bonded and 0 if the atoms are not bonded. The structures are converted from coordinates to adjacency matrix using *xyz2mol. Jensen (2020)* The assignment of bond/no bond is done using the xyz2mol program based on a simple extended Hückel theory (EHT) calculation and the Mulliken overlap population between each pair of atoms as implemented in RDKit (*Landrum, 2020*). The adjacency matrices for the endpoints of the IRC are compared with the adjacency matrices for the intended reactant and product structures to determine if a TS for the intended reaction is found. If the adjacency matrices of the IRC endpoint structures do not match those of the input reactant and product structures it may be due to the IRC not having completed as the IRC calculations often terminate before converging to reactant/product structures. Thus, the endpoints of the IRC are geometry optimized, and these structures checked by the same procedure. If either sets of structures (based on adjacency matrices) match the input structures, the TS for the given reaction is concluded to have been found and the search procedure terminated.

If the IRC did not result in a path connecting the input reactant and product, a constrained optimization on the TS guess, obtained as the maximum energy structure of the interpolated structures, is performed. The bond constraints are set up automatically by considering the difference in adjacency matrices of input reactant and product structure, resulting in a set of bonds being formed/broken during the reaction. only connectivity changes are considered, meaning that, e.g., going from a double bond to a single bond is not considered bond breaking. The length of the set of bonds are fixed to the values in the guess structure from the interpolation, and the remaining structure relaxed. The new TS guess is taken through the same procedure with TS optimization, IRC and check. If the TS is still not found, the entire procedure is repeated but using an electronic temperature of 6000 K in the RMSD-PP reaction path step.

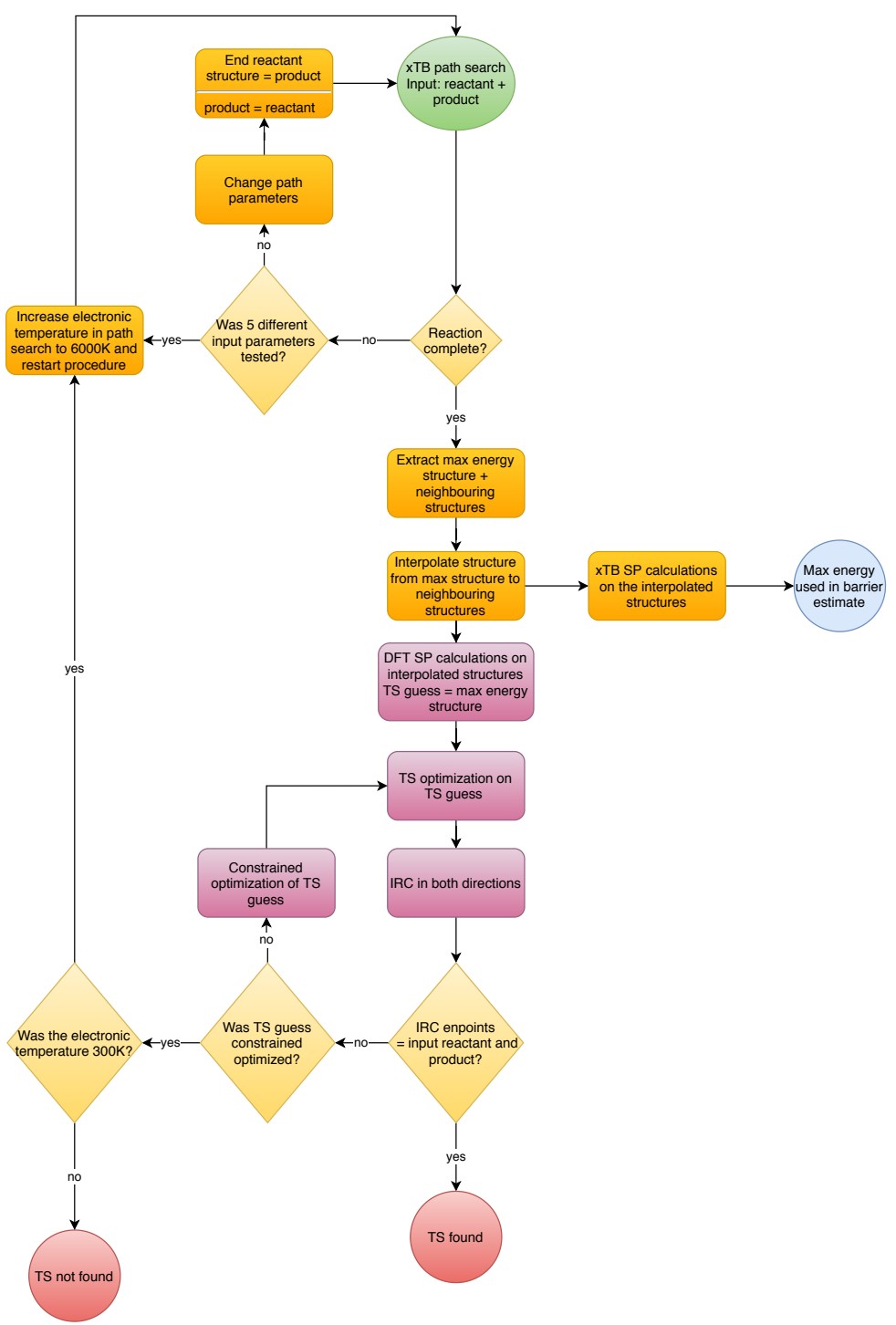

**Figure 1** **Flowchart describing the automated workflow implemented.** Orange steps depend solely on GFN2-xTB calculations, while purple steps rely on DFT calculations.

## Dataset

To test the TS localizer protocol, a preexisting data set from the literature is chosen to avoid bias in the choice of reactions studied. The data set used by Zimmerman to test his double-ended growing string method (GSM), consisting of 105 elementary reactions is used (*Zimmerman, 2013b*; *Zimmerman, 2013a*). Only reactions of neutral molecules and reactions where bond breaking/formation take place are included (i.e., excluding conformational changes). Thus, the test set consists of 100 elementary reactions including both simple and complicated reactions with between 1 and 6 bond changes (Table S3). To be able to use the TSs located by Zimmerman, the same level of theory for the DFT part of the procedure is used: UB3LYP/6-31G** (*Becke, 1988*; *Lee, Yang & Parr, 1988*; *Becke, 1993*; *Ditchfield, Hehre & Pople, 1971*; *Hehre, Ditchfield & Pople, 1972*). This level of theory lacks dispersion corrections, which are included in GFN2-xTB. This issue is unlikely to have any impact for the small compounds used in this study. However, for large compounds this method should be used in conjunction with dispersion corrected DFT.

All reactant and product structures were reoptimised using GFN2-xTB to verify that the structures have corresponding minima on the GFN2-xTB potential energy surface. This is the case for all reactions but reaction 16, as discussed further below. The DFT geometries for the reactant and product are used as input for the procedure described above.

## Approximate TS validation procedures

A popular approach in automated TS procedures is to either skip the IRC step and use alternative validation procedures for the TS or first screen the TS with alternative validation procedures before doing the IRC in an effort to save computational time (*Jacobson et al., 2017*; *Grambow, Pattanaik & Green, 2020*; *Zimmerman, 2013a*). Though the TS validation here is based on the IRC path and whether it connects the reactant and product, some of these alternative approaches are also tested. In particular, the TS vetting requirements suggested by *Jacobson et al. (2017)* are tested. The three requirements are: (1) There should be exactly 1 imaginary frequency of the Hessian, (2) at least one of the active bonds (bonds being broken or formed during the reaction) should have an intermediate length, and (3) that the eigenvector corresponding to the imaginary frequency should have motion along at least one of the active bond stretching modes. We use the same cutoff values for when a bond length is considered intermediate and when it is considered that the eigenvector has motion along a bond stretching mode as in the original article, that is: A bond length $r_{ij}$ between atom $i$ and $j$ is considered intermediate if

$$1.2 \leq \frac{r_{ij}}{r_i^{cov} + r_j^{cov}} \leq 1.7 \tag{2}$$

where $r_i^{cov}$ is the covalent radius of atom $i$ (*Landrum, 2020*). The eigenvector corresponding to the imaginary frequency, $v^{TS}$ is considered to move along the stretching mode of bond $i$, $v_i^{stretch}$ (unit vector), if the absolute value of the scalar projection of $v^{TS}$ on $v_i^{stretch}$ is larger than 0.33:

$$|v_i^{stretch} \cdot v^{TS}| \geq 0.33 \tag{3}$$

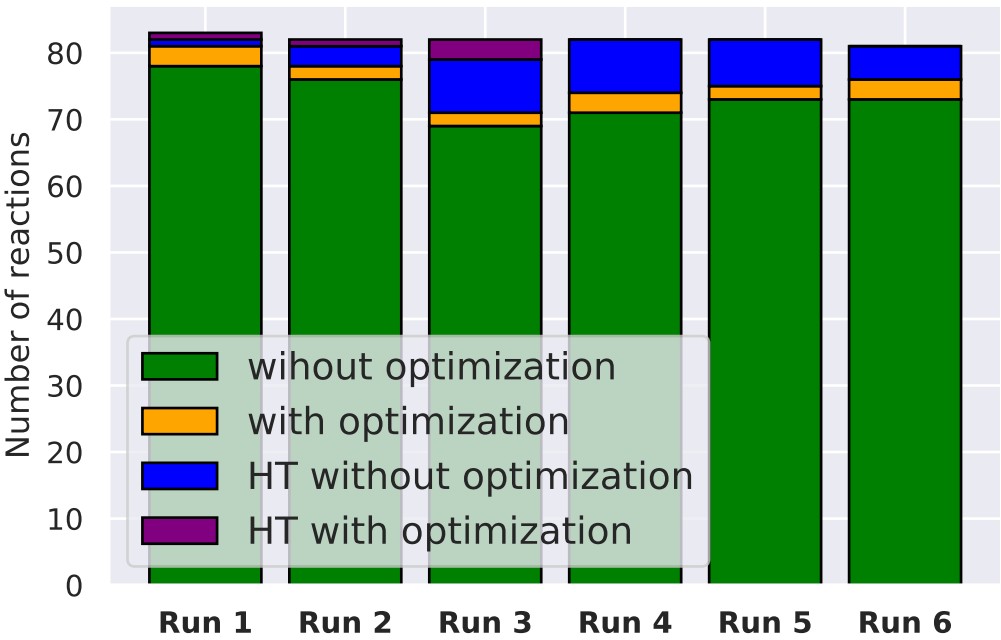

**Figure 2** **Distribution of the successful TSs localized for each of the 6 runs.** The GFN2-xTB TS guess structure is first used directly (without optimization) in a UB3LYP/6-31G** TS optimization. If that fails to find the TS, a constraint optimization is done and the TS optimization tried again. Finally, for the failed searches, the entire procedure is run at a higher electronic temperature (HT) of 6,000 K. Run 1–3 is done with parameter set 1 and Run 4–6 done with parameter set 2 (SI).

## RESULTS AND DISCUSSION

### Success rate

For each of the 100 reactions, the procedure is run three times with two different but similar parameter strategies for the GFN2-xTB path calculations (Table S2) for a total of 6 runs. The reason for running three times per parameter set is that the RMSD-PP procedure includes a random "initial distortion parameter" which can lead to slightly different reaction paths for each run.

Figure 2 shows the distribution of success rates for each of the 6 runs. Run 1–3 are with the same parameters (parameter set 1 in Table S2) and run 4–6 are with the same parameters (parameter set 2 in Table S2). The parameter sets are almost identical, the only difference is that the first run in parameter set 2 is initiated with a smaller push strength. The total number of successes is quite similar within the 6 runs (ranging between 81 and 83 TSs located) and the majority of the TSs are located using the guess structure from the RMSD-PP path directly. Combining all TSs located during the 6 runs, a total of 89 TSs are found. For the first parameter set (run 1–3) 85 TSs are located and for the second parameter set (run 4–6) 88 of the TSs are located. It is possible, that exploring a larger part of the parameter space allows localization of the last reactions.

For the reactions not located by the procedure (reactions 6, 10, 11, 16, 20, 35, 54, 68, 84, 90, and 96), the TS structures proposed by Zimmerman were further analysed.

**Figure 3** **The two reactions not found by the procedure.** Bonds broken in the reaction are indicated in red, bonds formed in blue.

However, they were first put through the same IRC validation procedure (with and without reoptimization of the TS). Only two of the remaining 11 reactions (reactions 16 and 84) went to minima corresponding to the proposed reactant and product structures, while the majority of the 9 reactions found an intermediate minimum structure along the way (Table S4), indicating that the reaction (at least for UB3LYP/6-31G**) is not an elementary reaction. The 9 reactions are not used in the following analysis, where the data set is now reduced to 91 reactions (89 of which the procedure managed to locate a TS for).

The two reactions, for which the TS search was unsuccessful, are shown in Fig. 3. The product in reaction 16 was the only structure that reacted when optimized with GFN2-xTB. After optimization the product became $NH_3 + BH_3 + NH_2BH_2$ and the product is thus not stable on the GFN2-xTB potential energy surface, which can affect the path optimization and thus the TS guess. During the DFT TS optimization the TS guess structure instead goes to the TS of reaction 9 (Table S3), which has a $\approx 8$ kcal/mol lower barrier than reaction 16. The other reaction not found, reaction 84, is a simple reaction and it is not clear why the TS of this reaction would be difficult to locate. Instead the TS of the reaction in Fig. 4 is found every time. Comparing the found TS with the true TS (Fig. 5) shows that the TSs are quite similar. The important difference seems to be the orientation of the methylene group in the middle.

## Comparison of xTB barrier estimates and DFT barriers

In this section we test whether the RMSD-PP reaction path can be used to distinguish reactions that have high and low barriers at the DFT level. If so, the RMSD-PP method could be used in the high throughput determination of reaction networks, where one is usually interested in relatively low-energy barriers. The 91 reactions for which a DFT TS is found, can be used to calculate the barrier of the reactions, which can be compared to the very cheap barrier estimates from the GFN2-xTB path. The barrier is calculated as the electronic energy of the TS (or maximum energy along the GFN2-xTB path) minus the electronic energy of the reactant. The reactant structures used were the same in both DFT

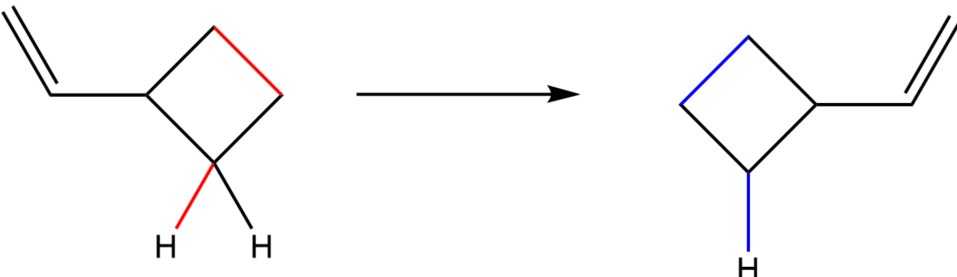

**Figure 4** **The reaction of the TS located when searching for the TS for reaction 84.** Bonds broken in the reaction are indicated in red, bonds formed in blue.

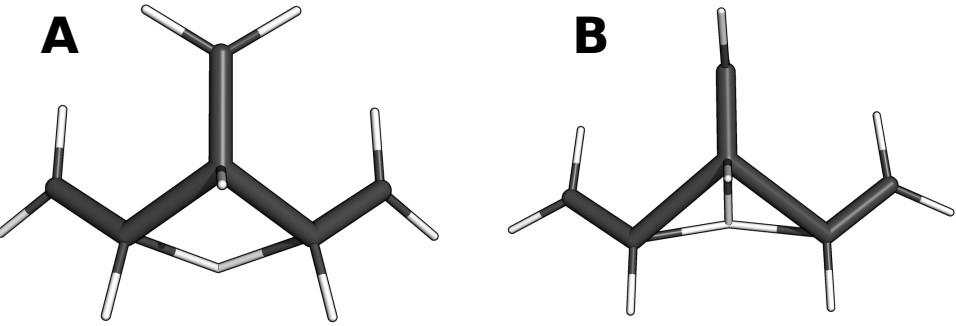

**Figure 5** **Comparison of the TS located (A) searching for the TS of reaction 84 (B).** A taken from the first run, coordinates for B from SI of the work of *Zimmerman (2013a)*.

and GFN2-xTB calculations (from *Zimmerman, 2013a*). This can affect the RMSD-PP barriers especially if either reactant or product structures are not stable on the GFN2-xTB surface as this can affect the path. All reactant and product structures were optimized with GFN2-xTB and only the product of reaction 16 changed bonding during optimization.

Part A, C and E of Fig. S1 show the correlation between the barrier estimated with GFN2-xTB and that calculated by DFT for the first parameter set (run 1–3). For each point is indicated the pull strength (color) and the push strength (size). Reactions, where the search was unsuccessful are labelled with red edges. Similarly, part B, D and F of Fig. S1 show the GFN2-xTB barrier estimate vs. DFT barriers for the three runs with parameter set 2 (run 4–6). As one would expect, higher pull and push values are needed for higher barriers. The mean absolute error (MAE) is between 14.9 and 19.2 kcal/mol for all six runs, and there is a wide spread of values and several outliers. So, generally speaking, the GFN2-xTB barrier from the RMSD-PP reaction path is a poor estimate of DFT barrier heights. However, in many reaction network studies the goal is to identify reactions that proceed at measurable rates at room temperature, which translates into barrier heights of no more than 30 kcal/mol. The correlation between GFN2-xTB and DFT is considerably better for these reactions. Figure 6A shows GFN2-xTB barrier estimates for all 91 reactions for run 1, while Fig. 6B includes only reactions with a DFT barrier of less than 30 kcal/mol.

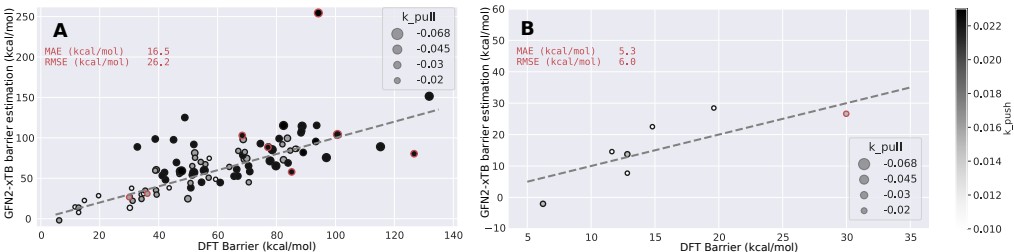

**Figure 6** **Barrier estimates from GFN2-xTB compared to DFT (UB3LYP/6-31G\*\*) barriers for the first run shown in Fig. 2.** $k_{pull}$ and $k_{push}$ values are given in Hartree per atom. For each point is indicated the pull strength (color) and the push strength (size). Reactions where the search was unsuccessful are labelled with red edges. (A) of the figure shows estimates for all (91) reactions while (B) shows barrier estimates for reactions with a barrier of less than 30 kcal/mol.

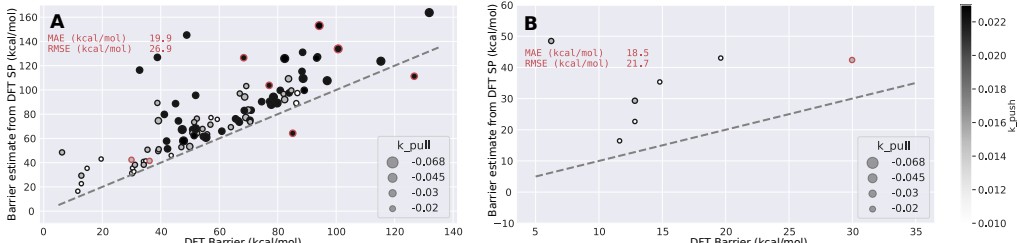

**Figure 7** **Barrier estimates from DFT single point energy evaluations compared to DFT (UB3LYP/6-31G\*\*) barriers for the first run shown in Fig. 2.** $k_{pull}$ and $k_{push}$ values are given in Hartree per atom. For each point is indicated the pull strength (color) and the push strength (size). Reactions where the search was unsuccessful are labelled with red edges. (A) of the figure shows estimates for all (91) reactions while (B) shows barrier estimates for reactions with a barrier of less than 30 kcal/mol.

For these 7 reactions the MAE is lowered to 5.3 kcal/mol (the MAE is 5.5 kcal/mol for the 7 reactions when calculated including all 6 runs). Reactions where the GFN2-xTB barrier is less than 40 kcal/mol, includes all seven reactions with DFT barriers less than 30 kcal/mol, in addition to 14–20 false positives (14, 17, 20, 14, 17, and 15 false positives for runs 1–6) where the DFT barrier is higher than 30 kcal/mol. If one excludes points where the absolute pull values are higher than 0.03 then the number of false positives drops to 11–14 (12, 11, 14, 11, 12 and 11 false positives for runs 1–6).

Recomputing the barriers using DFT single point calculations (Fig. 7, Fig. S2) leads to a slightly better correlation for the higher barriers but a worse correlation for the lower barriers (MAE=17.8 kcal/mol including all 6 runs), so the GFN2-xTB barriers are actually more useful.

## TS validation procedure

Here we test the performance of the validation procedures described in the Methods section: (1) exactly 1 imaginary frequency of the Hessian, (2) at least one of the active bonds (bonds being broken or formed during the reaction) has an intermediate length, and (3) the eigenvector corresponding to the imaginary frequency has motion along at least
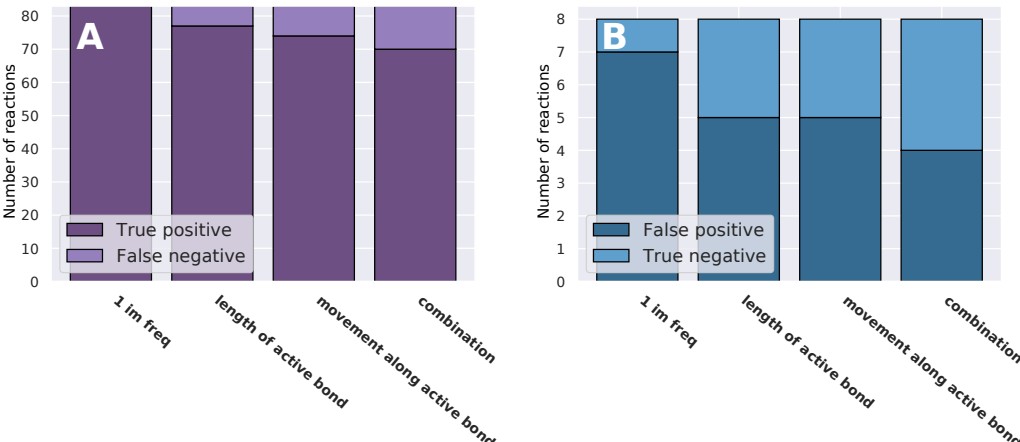

**Figure 8** Test of different TS validation methods for the (A) True TSs; and (B) wrong TSs of run 1.

one of the active bond stretching modes. The tests are applied to both the correct (83) and incorrect (8) TSs located during run 1. For the incorrect TSs, the first TS found (without constrained optimization) is used in the analysis. The outcome of the individual tests along with the combination of all three tests is shown in Fig. 8A for the found transition states of run 1 and in Fig. 8B for the failed transition states of run 1.

An effective validation procedure should discard as many wrong TSs as possible while not removing true transition states. The requirement, that the found transition state should have exactly 1 imaginary frequency is fulfilled for all 83 found TSs, but is also fulfilled for all but 1 (TS optimization failed) of the wrong transition states. Though the requirement can be applied without fear of throwing away true TSs, it is not very effective in filtering out wrong TSs. The requirement, that the TS structure should have at least one of the active bonds at an intermediate distance is fulfilled for 77 out of 83 true transition states and not fulfilled for three out of eight wrong transition states. Thus, applying this validation test to the transition state structures would have resulted in six correct TSs being filtered out. The last validation test, that the displacement vector of the imaginary frequency should be along at least one of the active bonds given the cutoff value above, is not fulfilled for nine of the transition states confirmed to be true by an IRC. Requiring all three validation tests to be fulfilled would have resulted in 13 of the 83 true transition states to have been filtered out. Four out of eight of the wrong transition states would also have been filtered out, but one needs to be very careful when applying these alternative validation tests, considering whether the saved computational time is worth more than the wrongly rejected transition states.

## SUMMARY

We present a method for the automatic determination of transition states (TSs) that is based on Grimme's RMSD-PP method (*Grimme, 2019*) for the rapid estimation of reaction paths using the GFN2-xTB semiempirical tight binding model (Fig. 1). The RMSD-PP method

estimates a reaction path between reactants and products by a geometry optimisation using an energy function augmented by two Gaussian biasing potentials, one "pushes" the structure away from the reactant and the other "pulls" the structure towards the product. Our method starts with a series of RMSD-PP calculations with increasingly larger push and pull strengths until reaction completion. The additional structures near the highest point on the reaction path are generated by interpolation and used for DFT single points and the highest energy structure is then used as an initial guess for a TS search. Upon convergence the TS is tested by an IRC calculation and if the TS is found to be incorrect then the initial guess structure is reoptimised with key bond lengths constrained and used as an initial guess for a new TS search. If that fails, the *entire* procedure is repeated but using an electronic temperature of 6000 K for the RMSD-PP calculations.

The method is tested on 100 elementary reactions used previously by Zimmerman and co-workers (Table S3). *Zimmerman (2013b)* and *Zimmerman (2013a)* For each of the 100 reactions, the procedure is run three times with two different but similar parameter strategies for the GFN2-xTB path calculations (Table S1) for a total of 6 runs. Combining all TSs located during the six runs, a total of 89 TSs are found. Only two of the remaining 11 reactions (reactions 16 and 84) went to minima corresponding to the proposed reactant and product structures, while the majority of the 9 reactions found an intermediate minimum structure along the way, indicating that the reaction (at least for UB3LYP/6-31G\*\*) is not an elementary reaction. Thus our method failed for only two reactions (Fig. 3), where for one of them the product is not a stable structure on the GFN2-xTB potential energy surface.

Furthermore, we show that the RMSD-PP barrier is a good approximation for the corresponding DFT barrier for reactions with DFT barrier heights up to about 30 kcal/mol. Thus, RMSD-PP barrier heights, which can be computed at the cost of a single energy minimisation, can be used to quickly identify reactions with low barriers, although it will also produce some false positives.

Finally, we show that various tests of whether the correct TSs have been found, produce several false positives and false negatives and should be used with care.

### Funding
Maria H. Rasmussen is supported by a research grant (00022896) from VILLUM FONDEN. The funders had no role in study design, data collection and analysis, decision to publish, or preparation of the manuscript.

### Grant Disclosures
The following grant information was disclosed by the authors:
VILLUM FONDEN: 00022896.

### Competing Interests
Jan H. Jensen is an Academic Editor for PeerJ Physical Chemistry.

## Author Contributions

- Maria H. Rasmussen conceived and designed the experiments, performed the experiments, analyzed the data, performed the computation work, prepared figures and/or tables, authored or reviewed drafts of the paper, and approved the final draft.
- Jan H. Jensen conceived and designed the experiments, analyzed the data, authored or reviewed drafts of the paper, and approved the final draft.

## Data Availability

The code and data is available at https://github.com/jensengroup/RMSD_PP_TS and https://sid.erda.dk/sharelink/EPvv68fOTp, respectively.

## Supplemental Information

Supplemental information for this article can be found online at http://dx.doi.org/10.7717/peerj-pchem.15#supplemental-information.

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
