# Peer review of "Fast and automatic estimation of transition state structures using tight binding quantum chemical calculations"

_PeerJ Physical Chemistry, doi:10.7717/peerj-pchem.15_

## Round 0.1 · original submission · Minor Revisions

I am pleased to agree with the reviewers impression that the presented manuscript offers important high quality work. I also follow the reviewers suggestions for improvements and would ask for revising the manuscript according to their suggestions.

Please provide a point-by-point response/assessment of their comments to support the final editorial decision.

Reviewer 1 ·

Basic reporting

A) The article is well written, and mostly, citations to the directly related works are given. In the introduction, one should add a citation to H.B. Schlegel’s WIREs article (DOI: 10.1002/wcms.34), which gives a good overview over different TS optimization techniques. Furthermore, some of the TS optimization techniques mentioned therein might be worth citing as well – for sure the one that has been used in the DFT re-optimization using Gaussian should be cited here (not just the Gaussian suite itself).

B) The introduction itself could be longer in the sense that the importance for high throughput barrier calculations could be highlighted (kinetic model generation for example). Something like this is written at the bottom of p.5, (below “Comparison of xTB barrier estimates and DFT barriers”) and could be also given in the Introduction.

C) The color codings should be explained in the figure captions of figures 3 & 4.

Minor:
a) The abbreviation RMSD-PP is actually never introduced, i.e., that “PP” stands for push-pull. In the same context, I find “Grimme’s RMSD-PP semiempirical tight binding reaction path method” in the abstract too coarse, as there are two components here:
The path search mechanism based on an atomic RMSD-based bias potential and the underlying electronic structure method, which has been the semiempirical GFN2-xTB method in Grimme’s work as well as this one. It would not harm to be more explicit, like “RMSD-PP in combination with the semiempirical tight-binding method…”, since we could, in principle, also combine RMSD-PP with DFT.
b) It says “GFN-xTB […] models” in the summary, but the paper reports exclusively GFN2-xTB. Were other methods tested? If not, I would just use GFN2-xTB here as well.
c) The x-axes in Figure 6 and 7 should be more explicit (“DFT barrier” or so).
d)
p.1: There is a period after “Grimme” in the introduction that needs to be removed.
p.2: “haven” -> “having”
p.4: “TS structures proposed by Zimmerman was further analysed”, change “was” to “were”.

Experimental design

I consider this paper an important contribution to the computational chemistry community.
A) The “2-4 optimization steps without any bias” on p.1 need to be explained further. How many states are exactly taken or what is the criterion for 2 vs 4 steps. It could be emphasized that this is not a path optimization, but an unconstrained geometry optimization, is that right?

B) Is the interpolation (p.2) done in Cartesian coordinates? Depending on how far the next neighbor is, this could be a bad choice, maybe not for the systems in this benchmark here, but for rotating substituents (phenyl for example).

C) p.2: “the endpoints of the IRC are geometry optimized, and these structures checked by the same procedure”. This needs further clarification. Due to the computational cost, I am assuming that this means that the structures are all optimized at the GFN2-xTB level. If so: does one actually ever need the adjacency matrix scheme or couldn’t this reoptimization procedure replace that entirely?

D) p.4: the “random ‘initial distortion parameter’” needs further explanation. Why can this not be done more deterministic?

Optional:
E) Did the authors test their procedure in combination with GFN2-xTB and an implicit solvation model (like GBSA)? This may perform better for some of the charged species – this is just a suggestion.

Validity of the findings

The authors analyze their data quite well and give explanations for each reaction in the supporting information.

A) This is a point that needs further elaboration:
Figure 7 shows promising results (DFT singlepoints on the xTB geometries), but it is not discussed in the text at all. I don’t know why. Here it is also important to note what that means. Are xTB geometries used throughout or only for the TS estimate, it looks like the latter, since the deviations all indicate an overestimation of the barrier. Maybe using xTB geometries (for the minima) could help here. I think this deserves more attention as it might give access to fast screening (a DFT singlepoint is mostly bearable).

B) The discussion/methodology around Figure 8 could be elaborated a bit more. Particularly, how are “wrong TS” classified if they fulfill all the 3 criteria? For example, are they higher in energy than the ones presented by Zimmerman, or is there a possibility that they actually found a lower TS?

C) The systems are mostly too small for this to be relevant, but the authors should mention that there exists an difference in the included physics: GFN2-xTB contains a London dispersion correction, while the DFT level does not. I understand that for consistency with Zimmerman, a dispersion-devoid model was used. But the authors should clarify that this could expectedly(!) lead to differences between both levels of theory for larger systems.

D) Is the tool/workflow available to the community?

Optional:
E) For the TS geometries in Figure 5: Did the authors try to compute S^2 expectation values with UB3LYP on both geometries? It would be interesting to know if the DFT geometry is spin-contaminated or if the xTB geometry has related issues (GFN2-xTB always favors low-spin configurations). Maybe a simple S^2 expectation value on both geometries shows something here (maybe not).

Additional comments

The manuscript “Fast and automatic estimation of transition state structures using tight binding quantum chemical calculations (#50973)” by Rasmussen and Jensen elaborates on an earlier presented reaction path scheme and assesses its performance for transition state structure prediction using a previously presented benchmark set. The manuscript is well written. It needs, however, attention in a couple of aspects. I suggest publication after the addressing the above raised issues.

·

Basic reporting

2) Thank you for providing a repository with data at https://sid.erda.dk/sharelink/EPvv68fOTp. The Supporting Information statement indicates that these are “data resulting from this study”. However, I note that:
a) Only reactant and product structures are provided, whereas (approximate) transition state structures are the key data resulting from this study. Would it be possible to include those? Including both the TS geometries obtained with the method and those optimized with DFT would allow other researchers to properly evaluate the work and results.
b) It is not described whether the provided structures are DFT geometries obtained from Zimmerman’s work, or geometries reoptimized with GFN2-xTB by the authors of this work. Please add this important meta-data.
c) I note that only 77 reactant structures (*r.xyz) and 91 product structures (*p.xyz) are included in the reactant_product_structures.tar.gz archive. They include also the reactant and product structures for the 5 reactions from Zimmerman’s set that were not used in this study (reactions 94, 95 and 102-105). This means that many reactant and product structures are missing.

3) The results for the barrier height prediction are not very clearly presented/quantified.
a) The authors state that “the RMSD-PP barrier is a good approximation for the corresponding DFT barrier for reactions with DFT barrier heights up to about 30 kcal/mol”, but they provide little/no direct quantification of this. They do state “The correlation between xTB and DFT is considerably better for these reactions.”, so please include a measure for this correlation here.
b) The presentation of Figures 6 and 7 is not optimal. I assumed that the six plots are for the data obtained from the 6 runs, with Runs 1-3 (obtained with parameter set 1) shown on one side, and Runs 4-6 (obtained with parameter set 2) on the other. This is probably the case (as indicated in the text), but the Figure legend itself states: “Figures (a), (d), and (e) correspond to runs 1, 2, and 3”, and it is not clarified what panels b,c,f show. This would be made much clearer by adding labels of “Run 1” etc. above the individual panels. Further, axis titles should be clarified, e.g. “Barrier xTB” --> “Barrier estimate from GFN2-xTB” and “Barrier” --> “DFT Barrier”. Further, the different ranges on the y-axis make visual comparison between runs difficult. Could the same range be employed throughout (e.g. 0-200), perhaps with outliers beyond 200 kcal/mole indicated using arrows?
c) The authors could consider placing the full Figure 6 and 7 in the Supporting Information (in which case point b) above could be ignored), and replacing them with plots for 1 run only, where they can show two plots for each: the total set of reactions as well as the reactions with DFT barriers below 30 kcal/mole only. This may help to better deliver a main message of the paper (“the RMSD-PP barrier is a good approximation for the corresponding DFT barrier for reactions with DFT barrier heights up to about 30 kcal/mol”, but not/less so for those with higher barriers). This will have the additional benefit of making the labels better readable (as they are currently quite small).

4) The presentation of the reactions is not very clear / may contain mistakes.
a) In the main text, Figures 3 and 4 would be much improved by including standard (electron pushing) arrows to describe the reaction. At least, the following should be added to the legend:
"Bonds broken in the reaction are indicated in red, bonds formed in blue".
b) Table S2 (and deposited data file reactions.txt) contain several identical reactions, i.e. reactants and products are identical (e.g. 1 and 2; 4,5 and 6; 10, 11, 14). This needs to be corrected. Further, the minus signs indicating atoms with formal negative charge are often difficult to spot. Can they be enlarged, or (ideally) be placed in circles (as in Figure 3)?

Experimental design

5) Some further clarification/information on the methods could be added.
a) I very much appreciate the short description of the RMSD-PP method, so that original paper(s) by Grimme describing the method do not have to be studied in detail. To clarify further how the method works, as relevant to the current manuscript, can you please describe: 1) How are points on the path defined (i.e. the 'distance' between them, and/or 2) how is it defined how many points there are on the full path?
b) Could there be a short indication/description of how the k_pull and k_push and alpha values used in the parameter sets were arrived at? Trial-and-error? Based on previous work?

Validity of the findings

1) The authors correctly describe their method as ‘fast’ in the title, however, exactly what this means is not described or reported clearly.
a) Table S5 indicates (for just two reactions) the wall-time required for TS estimation by their procedure, which is impressive. However, no reference is made to this table in the main text, nor is any approximate timing given for TS estimation. I suggest that the authors include somewhere (perhaps even in the abstract), that TS estimation can be obtained “within a few minutes on a single CPU”. Where such a statement is added to the text, Table S5 can be referred to. Further, it would be better if the CPU used for the Table S5 timings is described (e.g. speed in GHz, perhaps also type).
b) In the abstract and summary, the authors only describe that “RMSD-PP barrier heights [..] can be estimated at the cost of a single energy minimisation”. What this means is not very well defined, and is certainly hard to interpret by reading the abstract only. With RMSD-PP barrier height, to the authors mean barrier height estimation/calculation after TS estimation, or the whole procedure to obtain the estimate TS? With “single energy minimisation”, do the authors refer to a single energy calculation or a single structure optimisation, or the procedure of "start with the reactants, turn on push/pull potentials, minimize, and check, whether the system falls into products"? If energy calculation or optimisation, at what level of theory? This statement should be changed/clarified.

Additional comments

Rasmussen and Jenssen describe and evaluate a fast and automated procedure to determine transition state structures (TSs) for elementary reactions. The procedure is based on the RMSD-PP method from Grimme, together with his semiempirical tight-binding method GFN2-xTB. The work appears well-designed and conducted, and the methods and results are generally reported clearly and succinctly. I highly commend the authors for their efforts to produce a successful automated method for TS optimization that allows taking full advantage of the speed enhancements possible from Grimme’s methods, and making their code openly available and well-documented. There are, however, several things that should be clarified and some errors/omissions that should be corrected before the work can be accepted (see review sections).

The github repository is well structured and the code is extensively commented, which is highly appreciated. A suggestion: currently, the implementation can only be used with SLURM job scheduler. It would be nice to refactor the implementation to allow running the scripts with other job schedulers or at least directly on a local machine.

Finally, minor points regarding textual/language errors and small clarifications are annotated in the attached PDF. I would appreciate if the authors could address these too, to further improve the readability. In my view, it is not particularly useful for the community to make this annotated PDF available online; all main points raised are clearly described in the review text.

I hope this review will helpful to improve the presentation of your work.

Marc W. van der Kamp

---

## Round 0.2 · accepted · Accept

The authors have thoroughly addressed all comments by the referees and implemented updates to the manuscript. After checking the replies and changes I have no reservation accepting the manuscript without further review.